# Identification of miRNAs and Their Targets in *Cunninghamia lanceolata* Under Low Phosphorus Stress Based on Small RNA and Degradome Sequencing

**DOI:** 10.3390/ijms26083655

**Published:** 2025-04-12

**Authors:** Meng Li, Xiaopeng Ye, Ziyu Zhao, Yifan Zeng, Chaozhang Huang, Xiangqing Ma, Peng Shuai

**Affiliations:** 1College of Forestry, Fujian Agriculture and Forestry University, Fuzhou 350002, China; 1220428007@fafu.edu.cn (M.L.); 5220422013@fafu.edu.cn (X.Y.); 12304028024@fafu.edu.cn (Z.Z.); 52404022089@fafu.edu.cn (Y.Z.); chaozhangh@stu.fafu.edu.cn (C.H.); lxymxq@126.com (X.M.); 2Chinese Fir Engineering Research Center of National Forestry and Grassland Administration, Fuzhou 350002, China

**Keywords:** *Cunninghamia lanceolata*, miRNA, low phosphorus stress, degradome

## Abstract

Chinese fir (*Cunninghamia lanceolata* (Lamb.) Hook) is one of the main afforestation tree species in southern China. Continuous planting for multiple generations has led to a decrease in the content of available phosphorus in the soil. To adapt to low phosphorus stress, plants develop a series of physiological, biochemical, and developmental responses through self-regulation. Recent studies have shown that miRNAs play a regulatory role in plants’ responses to low phosphorus stress. However, the regulatory mechanism of miRNAs in Chinese fir in response to low phosphorus stress is still unclear. Here, we performed small RNA sequencing on the Chinese fir roots treated with normal phosphorus and low phosphorus and identified a total of 321 miRNAs, including 139 known miRNAs and 182 new miRNAs, with 43 differentially expressed miRNAs (DEMs). Integrative analysis combined with degradome sequencing data revealed that 193 miRNAs (98 known and 95 new) targeted 469 genes, among which 23 DEMs targeted 44 genes. Gene enrichment analysis indicated that under low phosphorus stress, transcription and transcriptional regulation, as well as signal transduction, were significantly activated in Chinese fir. Modules in the miRNA–target pathways, such as miR166/HD-ZIP III, miR169/NFYA7, miR529/SPL, and miR399/UBC23, may be the key regulatory factors in the response to low phosphorus stress in Chinese fir. In addition, we found that PC-3p-1033_8666 was significantly downregulated and that PC-5p-3786_2830 was significantly upregulated, which presumably respond to low phosphorus stress by indirectly affecting phosphorus-related hormone signaling or PSR genes. The identified miRNA–target network and significantly activated pathways in this study provide insights into the post-transcriptional regulatory mechanisms of Chinese fir adapting to low phosphorus environments, which can offer theoretical references for the stress resistance and superior variety breeding of Chinese fir.

## 1. Introduction

Chinese fir, an economically and ecologically important conifer, is one of the main afforestation species in southern China with the advantages of good wood quality and fast growth rate [1]. It not only plays an important role in timber production and ecological restoration but also contributes significantly to soil improvement and conservation of soil and water [2,3]. However, due to continuous multi-generation cultivation of Chinese fir plantations and improper fertilization in forest areas, soil fertility has decreased, leading to a reduction in the content of available phosphorus in the soil [4]. Phosphorus is one of the essential macroelements for plant growth, crucial for energy conversion, nucleic acid synthesis, protein phosphorylation, and the stability of cell membranes [5]. Plants take up phosphorus from soil in the form of inorganic phosphate (Pi), such as HPO_4_^2−^ or H_2_PO^4−^ [6]. However, Pi is one of the most underutilized nutrients due to its low diffusion rate, fixation with metals, and conversion to organic phosphorus through microorganisms [7]. Under low phosphate conditions, plants need to adapt to this restrictive environment by adjusting their physiological and molecular mechanisms to maintain survival and growth [8]. In recent years, with the advancement of molecular biology techniques, researchers have begun to explore in depth the mechanisms of plant responses to Pi starvation [9].

Over the course of evolution, plants have developed various strategies to cope with phosphorus deficiency stress. For example, a low phosphorus environment can stimulate the elongation and proliferation of lateral roots and root hairs, increasing the area for phosphorus absorption [10]. At the same time, the roots secrete more organic acids and enzymes, converting insoluble and unavailable phosphorus in the soil into soluble forms, thereby increasing the level of available phosphorus in the soil [11]. Additionally, the expression of phosphorus starvation response-related genes is activated, increasing the number and activity of phosphorus transporters, enhancing the uptake and transport of phosphorus for plants and then achieving efficient utilization of soil phosphorus resources [12]. Recent studies have shown that miRNAs, as an important class of signaling molecules, respond to low phosphorus stress and participate in the regulation of the expression of phosphorus starvation-related genes, consequently maintaining the phosphorus balance within the plant, and are the key regulatory molecules for plants to adapt to low phosphorus stress. MiR399 is the first miRNA molecule found to be involved in mediating low phosphorus signals [13]. MiR399 is induced by low phosphorus stress and regulates the expression levels and physiological functions of UBC24/PHO2 [14], which regulate the absorption and transport of phosphorus nutrition and maintain the phosphorus balance within the plant. In *Arabidopsis*, miR169 is inhibited by low phosphorus stress, while the expression level of its target gene NF-YA significantly increases [15]. MiR827 maintains the phosphorus balance of plants in a nitrogen-dependent manner, similar to miR399, and is specifically induced to be highly expressed under phosphorus deficiency [16]. MiR156/SPL, as an important signaling molecule, regulates plant growth and development and is also an important module for plants to adapt to low phosphorus stress [17]. With the discovery of the molecular mechanisms of miR399 and other miRNAs involved in regulating phosphorus balance, other miRNA families involved in phosphorus responses have also been identified through experiments, such as gene chips, high-throughput sequencing, and Northern blotting. Most plant miRNAs mediating phosphorus signal transduction have similar low phosphorus response characteristics, but some miRNAs have species-, tissue-, or organ-specific responses to low phosphorus. For example, the expression of miR395 in *Arabidopsis* seedlings is inhibited by low phosphorus stress [18], while it is induced and upregulated in white lupin [19]. The target genes of these miRNAs encode proteins with functional diversity: some target genes encode transcription factors that participate in the regulation of gene expression, while others encode factors responding to biotic or abiotic stress or enzymes (such as ubiquitin ligases or phosphatases). The latter can mediate post-translational modifications of proteins (such as ubiquitination, phosphorylation) or degradation pathways, and thus co-regulate the stress response process.

Current studies have identified 140 miRNAs from self-crossed families of Chinese fir through small RNA sequencing, of which 75.7% (106) were previously unrecorded, and 105 of these belonged to a new set with 6858 predicted target genes [20]. Cao et al. [21] studied the cytological changes, hormone levels, and dynamic gene expression during the processes of breaking primary dormancy and inducing secondary dormancy in Chinese fir seeds through transcriptome and degradome sequencing combined with PAREsnip. They identified 19 miRNAs, by which mRNA-targeted degradation played a key post-transcriptional regulatory role in the process of breaking primary dormancy in Chinese fir seeds. In addition, other studies have conducted small RNA sequencing on Chinese fir seeds, seedlings, leaves, stems, and callus tissues, identifying 115 conserved miRNAs and one novel miRNA, as well as elucidating the miR390-TAS3-ARF regulatory pathway [22]. Small RNA sequencing of the vascular cambium in Chinese fir identified 20 known miRNA families and 18 novel miRNAs, and it was found that miR156 and miR172 played a key role in regulating the transition of the vascular cambium from dormancy to active growth [23]. Both of these studies verified the corresponding target genes through 5’RACE experiments, few studies combine small RNA sequencing with degradome sequencing to reveal the regulatory mechanisms of miRNAs and their target genes in Chinese fir. This study identified miRNAs and their target genes in Chinese fir roots under low phosphorus stress through small RNA sequencing combined with degradome sequencing and analyzed the miRNA regulatory network. These studies not only help to reveal the molecular mechanisms of phosphorus stress response but also may provide new strategies to improve the phosphorus use efficiency of Chinese fir and other crops. Therefore, in-depth research on the molecular regulatory network of Chinese fir under low phosphorus stress is of great theoretical and practical value for enhancing plant productivity and sustainable agricultural development.

## 2. Results

### 2.1. Small RNA Sequencing and Analysis

In order to study the response of miRNAs in the roots of Chinese fir to Pi treatment, six small RNA libraries were constructed (two different treatments × three viological replicates) and sequenced using the Solexa technique [24]. The quality control analysis of the small RNA sequencing data revealed that the base quality value (*Q* value) was greater than 30, indicating good sequencing quality, which laid the foundation for subsequent data analysis. Subsequently, we performed a statistical analysis of the raw sequencing data output, obtaining the unique sequences of the sequencing data and the corresponding copy number of each unique sequence. At the same time, we compared the obtained sequences with the mRNA, RFam, and Repbase databases for filtering. Ultimately, 73,077,963 total reads (ranging from 7,227,792 to 16,184,397 per sample) and 16,805,320 unique reads (ranging from 2,095,238 to 3,269,733 per sample) were obtained (Table 1). We further conducted a statistical analysis of the total number (total) and the number of species (unique) of miRNAs in the filtered valid data based on their length distribution. Most of the data were distributed in the 20–24 nt range with a major peak at 21 nt (Figure 1a,b), which was consistent with the typical characteristics of Dicer enzyme cleavage, proving that the miRNA sequencing and effective sequence screening were successful.

### 2.2. Identification of Known and Novel miRNA

The clean reads were mapped to the precursors of specific species in miRBase 22.1 using a BLAST 2.2.25 search. Unique sequences that mapped to the hairpin arm or the complementary arm of mature miRNAs of specific species were identified as known miRNAs. Unmapped sequences were subjected to secondary structure prediction using the mfold software (http://www.unafold.org/mfold/applications/rna-folding-form-v2.php accessed on 7 April 2025). All sRNAs that could fold into a secondary structure were considered potential novel miRNA candidates. In total, 321 miRNAs were found in Chinese fir, including 139 known miRNAs and 182 novel miRNAs, and they were categorized into 4 different groups based on their relationship with the miRNA precursor sequences reported in the miRBase database. Among them, gp1, gp2, and gp3 were related to the miRNA precursor sequences reported in the miRase database, while gp4 was unrelated and represented entirely novel miRNAs (Appendix A). The identified miRNAs belonged to 205 precursor sequences, among which 61 precursor sequences each produced 2 mature miRNAs, 8 precursor sequences each produced 3 or more mature miRNAs, and the remaining 136 each produced 1 mature miRNA with 27 precursor sequences producing 27 miRNA–miRNA* pairs. The lengths of these mature miRNAs ranged from 18 to 25 nt, with both known and novel miRNAs predominantly concentrated at 21 nt (Appendix A). Based on calculations using Mfold, the free energy (dG) of the novel miRNAs ranged from −169.2 to −27.9 kcal/mol. The minimal folding free energy (MFE) was related to the sequence length, and the minimal folding free energy index (MFEI) provided a standard for comparing the MFE of miRNA precursors of different lengths in the roots of Chinese fir. The MFEI of novel miRNAs in the roots of Chinese fir ranged from 0.9 to 1.8 kcal/mol (Appendix A), which was conducive to forming stable hairpin structures.

Dicer enzymes and Dicer-like (DCL) enzymes exhibit a bias for the 5′ end base uridine (U) when recognizing and cleaving miRNA precursors, and the AGO1 protein also has a greater affinity for miRNA sequences with a 5′ end U base. Analysis of the base bias for total miRNA, known miRNA, and novel miRNA revealed that miRNA sequences of different lengths exhibited distinct base bias (Figure 2a–c). The first base at the 5′ end of both total miRNA and known miRNA significantly targeted the U base (Figure 2d,e), which is consistent with the miRNA preferential cleavage phenomenon reported in most plant species, indicating that our miRNA identification results were reliable. In contrast, the first base at the 5′ end of the novel miRNAs (18 bp and 19 bp) showed a strong base bias for the A base (Figure 2f).

To identify the conservation relationships of miRNAs in the Chinese fir roots with those in other species, the mature sequences of miRNAs from the Chinese fir roots were compared with the mature sequences of miRNAs from other species in the miRbase 22.1 database. It was found that the known miRNAs belonged to 50 miRNA families, among which the MIR396, MIR166, and MIR482 families had the most miRNA members with counts of 16, 10, and 10, respectively (Figure 3). The conservation analysis revealed that the mature sequences of miRNAs from the Chinese fir roots were identified in 83 species, with a significant proportion found in *Glycine max* (gma), *Picea abies* (pab), *Malus domestica* (mdm), *Populus trichocarpa* (ptc), and *Manihot esculenta* (mes) (Appendix A), indicating a high degree of conservation of plant miRNAs in evolution.

### 2.3. Differentially Expressed miRNA

By comparing the read counts generated by high-throughput sequencing, we identified 43 differentially expressed miRNAs (DEMs) in the Chinese fir roots in response to phosphorus stress (*p* value ≤ 0.05), including 26 novel miRNAs and 17 known miRNAs (Appendix A). When applying a more stringent criterion of total expression abundance >10, 30 differentially expressed miRNAs were found across the 6 libraries. The heatmap of differentially expressed miRNAs was shown in Figure 4a, with 31 DEMs significantly upregulated and 12 DEMs significantly downregulated. MiRNA members of the same family exhibited similar expression profiles, for example, two members of both the miR482 family and miR169 family were significantly upregulated due to phosphorus treatment. To verify the reliability of the miRNA TPM expression data, we selected 9 low phosphorus-responsive miRNAs for RT-qPCR assays, and the results showed that the expression patterns of the selected miRNAs were similar to the TPM expression patterns of the miRNAs from the sequencing results (Figure 4b).

### 2.4. Target Genes Identification and Functional Enrichment Based on Degradome Sequencing

Analyzing the sequencing data from two degradome libraries (Dep_plus and Dep_min), a total of 39,016,365 and 40,529,349 raw reads were obtained, representing 9,111,132 and 9,732,496 unique raw reads, respectively. After removing low-quality sequences and the sequences with short 3′ adapter cutting site tags (unique reads < 15 nt), the number of unique mappable reads in the Dep_plus and Dep_min were 9,039,689 and 9,657,864, respectively. Statistical analysis of the coverage of mRNA and degradation fragments obtained from the degradome revealed that the unique transcript mapped reads in the 2 groups were 5,787,569 (63.52%) and 6,075,096 (62.42%), covering a total of 187,329 transcripts (accounting for 29.35% of the 638,227 input transcripts) (Table 2). Based on the characteristic abundance of each occupied transcript position, all potential cleaved transcripts were categorized into five categories (categories 0, 1, 2, 3, 4) [25], and categories 3 and 4 were excluded to improve prediction accuracy. These miRNA cleavage sites were represented in the form of target plots (T-plots) as shown in Figure 5. Using degradome sequencing, we were able to identify 469 targets for 193 miRNAs (98 known and 95 novel). We also identified 23 DEMs targeting 44 genes, with target annotations showing that they primarily belonged to transcription factors (TFs), transport protein, binding protein, reductases, catalytic enzymes, and other genes (Appendix A).

To further understand the functions of these identified target genes, we conducted GO enrichment analysis. GO enrichment analysis of DEGs revealed that the most enriched biological processes included regulation of transcription, DNA-templated (GO:0006355), transcription, DNA-templated (GO:0006351), and defense response (GO:0006952). In terms of cellular components, the majority of target genes were enriched in nucleus (GO:0005634), cytoplasm (GO:0005737), and integral component of membrane (GO:0016021). In molecular functions, the top three most represented were transcription factor activity, sequence-specific DNA binding (GO:0003700), DNA binding (GO:0003677), and ADP binding (GO:0043531) (Appendix A). The analysis indicated that miRNA targets were concentrated in nuclear transcription factors and respond to stress. To investigate the association between transcription factors and low phosphorus responsive miRNAs in the GO analysis, network analysis was conducted using Cytoscape software v.3.10.0. The network construction included DEMs, transcription-related GO terms, and stress-responsive genes (Figure 6). These genes were closely related to plant growth and development as well as stress responses, indicating that miRNAs played an important regulatory role under low phosphorus stress. KEGG pathway analysis of the target genes of DEMs revealed significant enrichment in Aminoacyl-tRNA biosynthesis (ko00970), Plant hormone signal transduction (ko04075), and Plant–pathogen interaction (ko04626), suggesting that in response to changes in phosphorus conditions, plants could adapt to environmental changes by regulating internal hormones, protein synthesis, and even the formation of mycorrhizal hyphae (Appendix A).

### 2.5. miRNA Regulatory Network and Key Modules in Response to Low Phosphorus Stress

A DEM–target network was constructed to focus on the post-transcriptional regulation of Chinese fir in response to low phosphorus stress. Network analysis of the regulatory network indicated that 23 of 43 DEMs targeted 44 target genes, including 9 known miRNAs and 14 novel miRNAs (Figure 7, Appendix A). These genes belonged to transcription factors such as NFYA7, SPL, HD-ZIP III and others. The gene responding to low phosphorus stress, miR399 [26], targeted the ubiquitin-conjugating enzyme genes Cl_399 (UBC23). MiR169 [27] targeted 4 genes of NFYA7 family, which were annotated to the Aminoacyl-tRNA biosynthesis pathway. Other identified gene responding to low phosphorus stress, miR166 [27], also targeted 4 genes, which belonged to HD-ZIP III family. We found that miR529 primarily targeted SPL family genes, with a total of 3 miRNA–target pairs, indicating that the miR529-SPL played an important role in the regulatory network. In addition, 14 novel miRNAs target protein-coding genes, such as ABCC10, AGO, and ZAT, and TFs, including ARID2 and other genes. These miRNA–target pairs internally regulated plants, potentially adjusting hormone secretion and root growth to enhance the ability to absorb soil phosphorus.

## 3. Discussion

MicroRNAs play important roles in plant growth and development by regulating hormone secretion, signal transduction, and abiotic stress responses (such as temperature, salinity, drought, and heavy metal stress) [28,29,30,31,32]. However, the regulatory mechanisms of miRNAs under phosphorus stress are still rarely reported. Therefore, this study focuses on Chinese fir to explore the regulatory mechanisms of root miRNAs under low phosphorus stress. Currently, there are 38,589 hairpin precursor miRNAs and 48,860 mature miRNA sequences registered in the species of the miRBase database, including *Arabidopsis thaliana* (326 precursors, 430 mature forms), *Oryza sativa* (604 precursors, 757 mature forms), and *Glycine max* (684 precursors, 756 mature forms), while Chinese fir has only 4 precursors and 4 mature records (miRBase, release 22.1) [33]. With the emergence and development of high-throughput sequencing, different numbers and types of miRNAs have been successively discovered in most plants, such as *Zea mays* [34], *Triticum aestivum* [35], and *Luffa cylindrica* [36]. In this study, based on the sRNA-seq data of Chinese fir roots, 321 miRNAs were identified, including 139 known miRNAs belonging to 50 highly conserved miRNA families, and 182 novel miRNAs (Appendix A). Statistical analysis found that 46 miRNAs in this study have been reported in previous studies (Table 3). They identified different numbers and types of miRNAs in multiple tissues [22], vascular cambium [23], seeds [21], and stems and leaves [20] of Chinese fir (Appendix A). Venn analysis revealed five overlapping miRNA families (MIR156, MIR164, MIR166, MIR171, and MIR396) in all known Chinese fir miRNAs (Appendix A). Notably, 22 miRNA families were discovered in this study that had not been previously recorded in Chinese fir, and the number of new miRNAs was also higher compared to other reports. Compared with previous studies, the systematic mining of this study not only revealed the impact of tissue specificity on miRNA expression profiles but also provided new resources for parsing the genetic regulatory network of Chinese fir. In addition, the miRNA length distribution was mainly characterized by a peak at 21 nt (Figure 1), which was highly consistent with the characteristics of gymnosperms, such as Masson pine (*Pinus massoniana*) [37], Norway spruce (*Picea abies*) [38], and Japanese cedar (*Cryptomeria japonica*) [39], further supporting the reliability of the data.

With the widespread application of high-throughput sequencing technology, the dynamic regulatory mechanisms of miRNAs in plants under low phosphorus stress have gradually been revealed. Existing studies have shown that 45 and 55 differentially expressed miRNAs were identified in the roots and shoots, respectively, of the model plant *Arabidopsis thaliana* under low phosphorus stress [40]. In *Medicago sativa*, 47 and 44 phosphorus-responsive miRNAs were found in the roots and shoots [41], and 26 and 18 phosphorus-responsive miRNAs were found in the roots and leaves of soybean [42]. In this study, a total of 43 miRNAs significantly responding to low phosphorus stress were identified, including 17 known miRNAs and 26 novel miRNAs, with a clear bias in their expression (31 significantly upregulated and only 12 downregulated). Further analysis revealed that families members of the MIR319, MIR167, and MIR482 shared functional conservation with previously reported low phosphorus-responsive miRNAs, such as MIR399 and MIR166. Through degradome sequencing, we successfully characterized 44 target genes of 23 DEMs. Functional annotations showed that these target genes mainly encoded transcription factors (such as SPL and NFYA), transporter proteins, and enzyme genes, which enhance plant adaptation to low phosphorus by regulating phosphorus starvation response (PSR), root system architecture remodeling, and hormone signaling pathways (such as auxin and ethylene). It was worth noting that the remaining 20 DEMs may participate in stress response through atypical regulatory mechanisms, such as inhibiting mRNA translation.

MiR399 is the first miRNA that has been proven to respond to low phosphorus stress [26]. In *Arabidopsis thaliana*, three miR399 target genes have been predicted: encoding a phosphorus transporter (PHT1;7), a DEAD-box helicase, and a ubiquitin-conjugating E2 enzyme, among which the E2 enzyme encoded by UBC24 has been verified. In this study, miR399 only targeted one gene (UBC23, which encodes the protein ubiquitin ligase E2 23). Liu et al. [42] found that the miR399 family could regulate the absorption and accumulation of phosphorus in soybean by controlling the target genes PHO2 and PT5, with PHO2 encoding the protein ubiquitin ligase E2. In this study, miR399d was significantly upregulated under low phosphorus stress, while its target gene UBC23 was upregulated, which was inconsistent with the conclusion pointed out by Bari et al. [43] that the expression levels of miR399 and its target gene in rice were negatively correlated. In addition, Wang et al. [44] found in their study on the molecular mechanism of maize adaptation to low phosphorus stress that ZmmiR399 was upregulated by low phosphorus induction, and its target gene ZmPHO2 was downregulated, releasing the phosphorus transporter PHO1 and PHT1s. Tao et al. [45] also found in their study on the regulation of soybean phosphorus balance and flowering by miR399 that the expression of soybean miR399 was induced by low phosphorus, and the expression level of its target gene GmPHO2-1 was negatively correlated with miR399. Research has shown that most miRNA–target regulation is mainly negative, with very few miRNAs able to positively regulate target genes [46]. However, the miR399 identified in this study belonged to the latter. This may be because miR399 indirectly affects the expression of target genes by influencing the metabolic pathways or hormone signaling of plants, indicating that target genes in plants may be regulated by multiple factors.

In addition to miR399, other genes identified in plants that respond to low phosphorus stress include miR166 and miR169. MiR166 and miR169 have been detected to respond to phosphorus limitation in the phloem sap of rapeseed, suggesting that plants automatically adjust under low phosphorus conditions [47]. Here, both miR166 and miR169 were significantly upregulated under low phosphorus induction. Through degradome sequencing and analysis using GO and KEGG, we found that Chinese fir miR166 targeted HD-ZIP III family genes (ATHB-15), and miR169 targeted NFYA7 genes. ATHB-15 and NFYA7 are related to transcription, regulation of transcription and lateral root formation. It has been reported that under low phosphorus treatment of wheat, tae-miR169 regulated TaNFYA1-6B1, promoting the root growth of wheat, which was beneficial for efficient phosphorus absorption and increasing the number of spikes and grain yield of wheat [48]. Sha et al. [49] found that miR166a and miR166b were significantly upregulated in the low phosphorus-induced bud reservoir, and miR166 targeted the HD-ZIP transcription factor in their study of soybean miRNA response to phosphorus deficiency. HD-ZIP proteins can affect plant root development and hormone signaling, indirectly influencing plants’ response to low phosphorus stress [50]. These results indicated that miR169 and miR166 play a key role in the process of plant phosphorus absorption by post-transcriptionally regulating target genes, and by finely regulating the expression and activity of phosphate transporters, plants can adapt to different phosphorus environments. In addition, under low phosphorus conditions, plants can induce the expression of downstream PSR (phosphate starvation response) genes through the synergistic action of multiple transcription factors and respond systemically to low phosphorus stress by integrating transcription, post-transcription, protein, mycorrhizal symbiosis, and root morphological changes. Kuo and Chiou [27] have found in their research progress on phosphorus-responsive miRNAs that the miR166 and miR169 families universally respond to low phosphorus stress in different plant species, which may synergistically participate in a conserved phosphorus signaling pathway in plants.

Under low phosphorus stress, plants may affect cell functions through various mechanisms, including influencing signal transduction, energy metabolism, gene expression, and membrane stability. Our study found that in Chinese fir, the differentially highly expressed miR529 targeted the SPL gene involved in plant hormone signaling, which also played an important role in the regulatory network. The SPL gene family is widely involved in the regulation of growth, development, and stress responses in plants. For example, in *Arabidopsis thaliana*, the SPL1 gene was upregulated under low phosphorus conditions, participating in regulating rhizosphere acidification reactions, thereby helping plants adapt to low phosphorus environments [51]. Studies have found that under low phosphorus stress, miR156 and its target gene SPL9 were involved in a series of low phosphorus response processes in plants, regulating the downstream miR399f to form a signal pathway in *Arabidopsis* that responded to low phosphorus [52]. In addition, we found that three known miRNAs (pab-miR319a_L+2R-1, atr-MIR393-p5_2ss14TA17AG, and gma-miR6300_R+7) and five novel miRNAs (PC-3p-7552_1533, PC-3p-507_14846, PC-3p-4864_2279, PC-3p-1033_8666, and PC-5p-3786_2830) had high expression levels and were differentially expressed under low phosphorus stress, which meant these miRNAs played an important role in Chinese fir’s adaptation to low phosphorus stress. Among them, PC-3p-507_14846 targeted an unknown protein, while the two target genes of PC-3p-1033_8666 were ABCC10 and AGO7. There was currently no direct evidence indicating that these proteins had a clear relationship with low phosphorus stress. They may indirectly affect the response to low phosphorus stress by promoting transmembrane transport of substances and reducing the degradation of PSR by miRNA. The RGA1 gene targeted by PC-5p-3786_2830 may regulate plant growth and development and help plants adapt to low phosphorus environments by regulating hormone signaling and interacting with phosphorus metabolism-related genes. Studies have found that under low phosphorus stress, OsRGA1 positively regulated the growth of rice root systems [53]. Compared with conventional phosphorus treatment, low phosphorus stress significantly promoted the growth of root systems in rice seedlings of various genotypes. Moreover, the increase in root system growth in the overexpression line ROE2 (RGA1 Overexpression 2) was greater than that in the wild type, indicating that OsRGA1 has a positive regulatory effect on root system growth under low phosphorus stress.

## 4. Materials and Methods

### 4.1. Plant Materials and RNA Extraction

The “061” clone of Chinese fir seedlings from Fujian Yangkou State-owned Forest Farm (117°9′ E, 26°82′ N), with a height of about 25 cm and in good health, were selected. We removed the light substrate and removed the sea sand. And then we divided them into normal and low phosphorus groups, with 8–10 seedlings each for sand cultivation, and allowed a 7-day recovery period. We prepared a 1/3 strength Hoagland nutrient solution (1 mmol/L KH_2_PO_4_ for normal, 0 mmol/L KH_2_PO_4_ for low phosphorus with equivalent amounts of KCl supplementing potassium), watered each plant with 100 mL of nutrient solution every 2 days, and treated the 2 groups for 20 days. Finally, we sampled the roots of Chinese fir seedlings after treatment, with three biological replicates for each treatment, which were immediately frozen in liquid nitrogen and stored at −80 °C, and extracted total RNA using the RNeasy^®^ Plant Mini Kit from the TIANGEN company. RNA quality and purity were checked using agarose gel electrophoresis and a NanoDrop 2000 spectrophotometer (Thermo, Wilmington, DE, USA) at 260/280 nm (ratio >2.0), and then, the samples were sequenced at LC-BIO (Hangzhou, China).

### 4.2. Small RNA Library Construction and Sequencing

The experimental process was carried out according to the standard protocols provided by Illumina, using the TruSeq Small RNA Sample Prep Kit (Illumina, San Diego, CA, USA) to construct sRNA libraries for 6 samples. After extracting total RNA from the samples, small RNAs within the range of 18–30 nt were isolated, and 5′ and 3′ adapters were ligated, followed by RT-PCR amplification. The PCR products were then purified through gel electrophoresis to obtain the sRNA libraries. Once library construction was completed, the prepared libraries were sequenced using the Illumina Hiseq2500 platform with a single-end 1 × 50 bp read length. The original sequencing data, in image format, was processed and optimized via base calling. Data with a Q30 (indicating a base error probability of less than 0.1% at a given position due to the sequencer) exceeding 80% was deemed of high quality. Subsequently, the quality-assessed data were filtered and aligned with databases, like Rfam and Repbase.

### 4.3. Filtering of Clean Reads and Identification of miRNAs

Raw reads were subjected to an in-house program of LC-BIO, ACGT101-miR (LC Sciences, Houston, TX, USA) to remove adapter dimers, junk, low complexity, common RNA families (rRNA, tRNA, snRNA, snoRNA), and repeats. The analysis process of the software was carried out according to the steps described by Wang et al. [54]. The retained sequences were aligned against various RNA database sequences (excluding miRNA), such as the mRNA database, the RFam database (including rRNA, tRNA, snRNA, snoRNA, etc.), and the Repbase database (a database of repetitive sequences), and filtered accordingly. The resulting data, which had passed these filters, were considered valid and used for subsequent small RNA data analysis.

Sequences of 18–25 nt were selected and aligned to the precursors of specific species in miRBase 22.1 via BLAST 2.2.25, allowing for length variations at the 3′ or 5′ ends and one internal mismatch. Sequences matching the stem arms where known mature miRNAs are located were defined as known miRNAs. Sequences matching the other stem arm of the corresponding precursor were considered as novel 3p or 5p-derived miRNA candidates. Unmatched sequences were further aligned to the precursors of other species in miRBase 22.1, and the matched precursors were subjected to genomic localization (BLAST alignment to the specific species’ genome). Both of the above categories were classified as known miRNAs. For the unmatched sequences, after genomic alignment, the flanking 120-nt sequences were extracted and the hairpin structures were predicted using RNAfold (http://rna.tbi.univie.ac.at/ accessed on 23 March 2025). Sequences with no homology were defined as potential novel miRNA precursors. Our laboratory provided the reference gene databases [55].

### 4.4. Differential Expression Analysis of miRNA

Normalized methods were employed to determine the expression profiles of miRNAs, and then, differential expression of miRNAs was analyzed by Student’s *t*-test. For samples with biological replicates, differentially expressed miRNAs (DEMs) were identified with a threshold of *p* ≤ 0.05.

MiRNA was extracted from the Chinese fir roots under low phosphorus and normal phosphorus conditions, and RT-qPCR was used to verify DEMs and analyze their expression patterns. The miRNA Real-Time PCR Assay kit from Beijing Aidelai Biological Technology Co., Ltd. was used for reverse transcription and RT-qPCR. The 5.8s rRNA gene was selected as the reference gene, and specific forward primers for each miRNA were designed according to their sequences (Appendix A), with the reverse primer being a universal primer provided by the kit. The program was run on the Applied Biosystems QuantStudio 1 Plus quantitative PCR instrument, and the relative expression of miRNA was calculated using the 2^−∆∆Ct^ method.

### 4.5. Degradome Sequencing, Target Identification and Analysis

Total RNA was extracted using Trizol reagent (Invitrogen, Carlsbad, CA, USA) following the manufacturer’s procedure. The total RNA quantity and purity were analysis of Bioanalyzer 2100 and RNA 6000 Nano Lab Chip Kit (Agilent, Santa Clara, CA, USA) with RIN number >7.0. Approximately 20 µg of total RNA was used to prepare the degradome library. The method followed that described by Addo-Quaye et al. [56] with some modification.

After adapter removal and low-quality filtering on the Illumina platform, the raw sequencing data were processed through the CleaveLand pipeline [56] to identify potential cleavage sites. The degradome sequencing reads were precisely aligned to the reference transcript database [55], and only perfectly matched sequences were selected for subsequent analysis. The t-signature sequences were reverse complemented and aligned with the miRNAs identified in this study. Targetfinder was used to predict the paired mRNAs, which were then integrated with the degradation density files to screen for common target genes (allowing ≤4 alignment scores). T-plots were constructed to visualize the target degradation patterns. The final target genes were subjected to homology alignment via BlastX (https://blast.ncbi.nlm.nih.gov/Blast.cgi?PROGRAM=blastx&PAGE_TYPE=BlastSearch&LINK_LOC=blasthome accessed on 7 April 2025), and then, GO and KEGG enrichment analyses were performed using ggplot2. The target annotation information of differentially expressed miRNAs from degradome sequencing was integrated, and Cytoscape software v3.10.0 was used to construct miRNA–gene–GO network diagrams and DEM–target regulatory networks.

## 5. Conclusions

This study discovered that miRNAs were involved in the low phosphorus stress response in Chinese fir. A total of 321 miRNAs were identified through small RNA sequencing, including 139 known miRNAs and 182 novel miRNAs. By combining degradome sequencing, we identified 469 targets for 193 miRNAs (98 known and 95 new), among which 23 DEMs targeted 44 genes. The comprehensive analysis of target enrichment and miRNA regulatory networks suggested that miR399/UBC23, miR166/HD-ZIP III, miR169/NFYA7, and miR529/SPL) may be key regulatory factors in the response of Chinese fir to low phosphorus stress. Additionally, the newly discovered PC-3p-1033_8666 and PC-5p-3786_2830 may indirectly participate in the response to low phosphorus stress by affecting genes or hormones associated with phosphorus metabolism. The intrinsic molecular mechanisms of plant absorption of phosphorus and response to low phosphorus stress revealed in this study can improve the efficiency of plant absorption of soil phosphorus and provide new clues for gene improvement breeding.

## Figures and Tables

**Figure 1 ijms-26-03655-f001:**
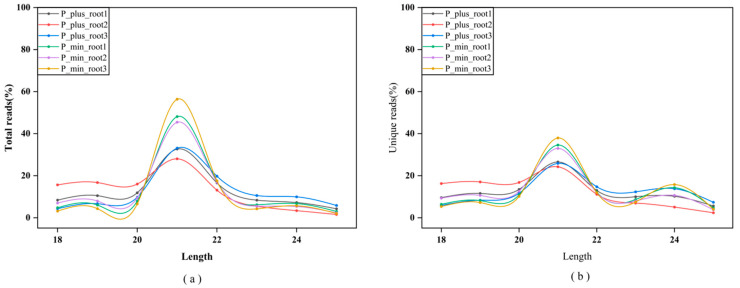
Length distribution of total (**a**) and unique (**b**) sequences in six sRNA libraries in Chinese fir.

**Figure 2 ijms-26-03655-f002:**
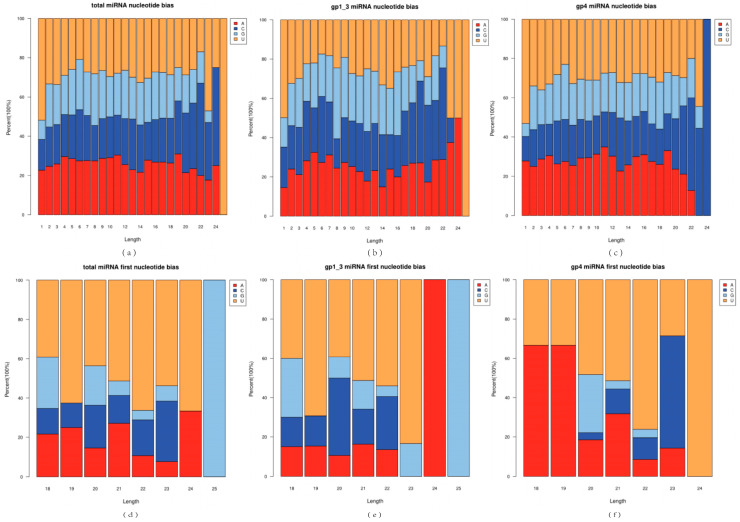
Base bias analysis of total miRNAs (**a**,**d**), known miRNAs (**b**,**e**), and novel miRNA (**c**,**f**). (**a**–**c**) show the base preferences at each position; (**d**–**f**) show the preferences for the first base of miRNA sequences of different compositions.

**Figure 3 ijms-26-03655-f003:**
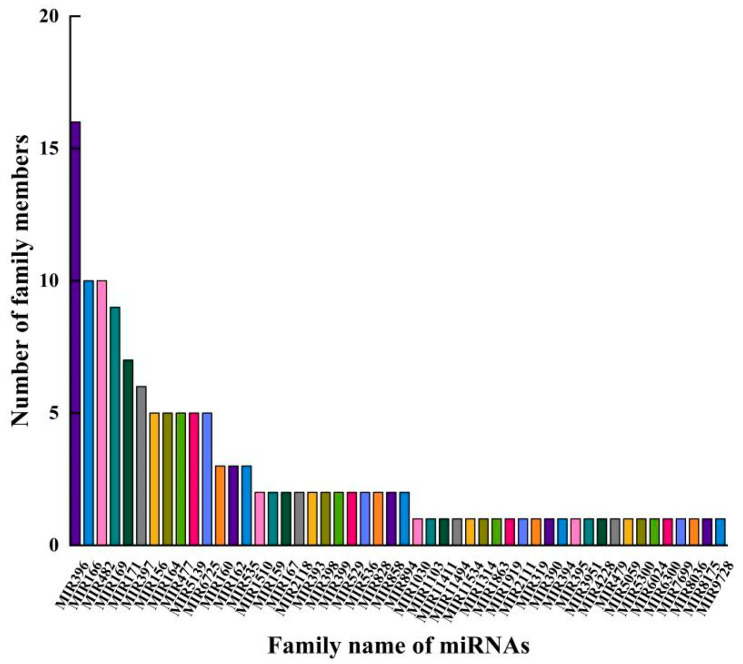
MiRNAs and their family numbers in Chinese fir root.

**Figure 4 ijms-26-03655-f004:**
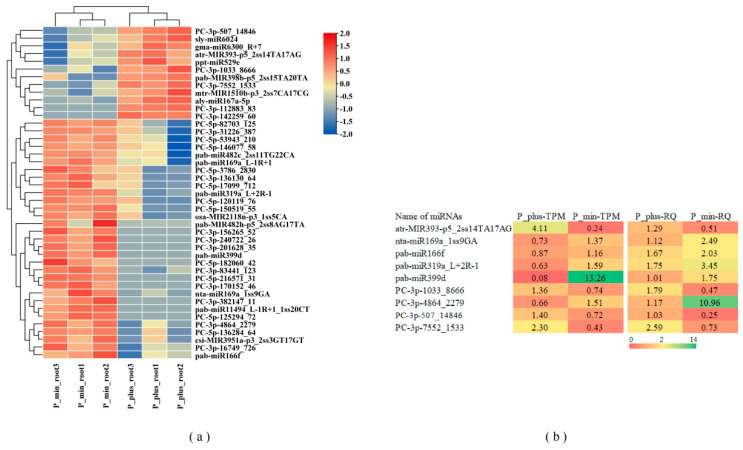
Phosphorus-responsive miRNAs in Chinese fir. (**a**). The expression profiles of DEMs. Red indicates higher levels of miRNA, and blue indicates lower levels of miRNA. (**b**). TPM, transcripts per million. RQ, relative expression level. The colors for TPM are based on log_2_ (fold change) values, and RQ is based on the relative expression levels from RT-qPCR.

**Figure 5 ijms-26-03655-f005:**
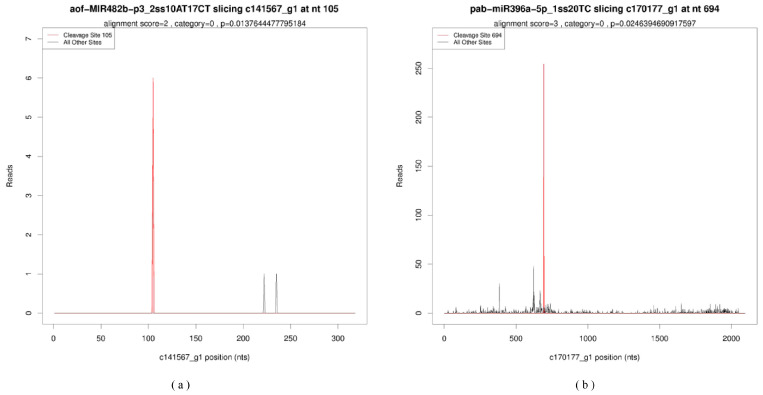
Target plots of the targets cleaved by the miRNAs. (**a**) Target plots of aof-MIR482b-p3_2ss10AT17CT. (**b**) Target plots of pab-miR396a-5p_1ss20TC.The *x*-axis represents the full length of the target gene sequence, and the *y*-axis represents the number of reads at a specific cleavage site. The red line represents the predicted cleavage site of the corresponding miRNA.

**Figure 6 ijms-26-03655-f006:**
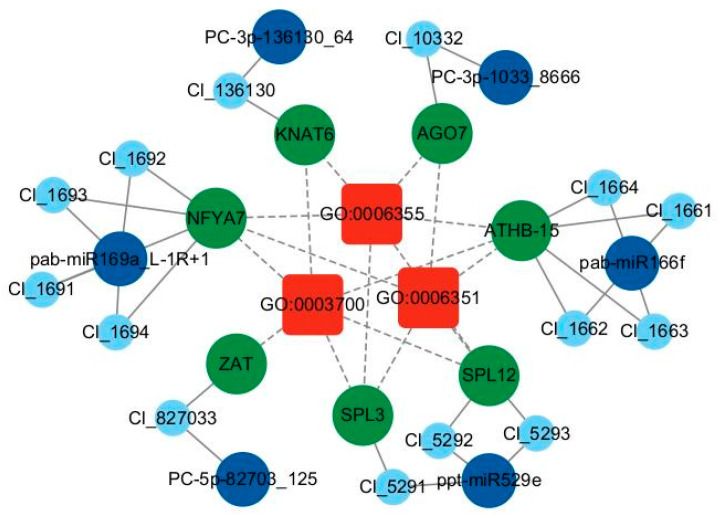
MiRNA–gene–GO network diagram. Red represents GO terms, green represents the symbols corresponding to target genes, dark blue represents DEMs, and light blue represents target genes. GO:0006351: transcription, DNA-templated. GO:0006355: regulation of transcription, DNA-templated. GO:0003700: transcription factor activity, sequence-specific DNA binding.

**Figure 7 ijms-26-03655-f007:**
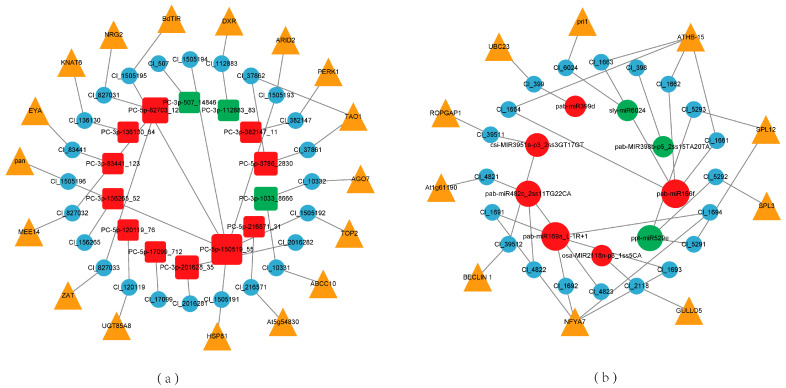
DEM–target regulatory network. (**a**) DEM–target regulatory network for novel miRNA. (**b**) DEM–target regulatory network for known miRNA. Red square nodes represent upregulated novel miRNAs, and green square nodes represent downregulated novel miRNAs. Red circular nodes represent upregulated known miRNAs, and green circular nodes represent downregulated known miRNAs. Blue circular nodes represent target genes. Orange triangle nodes represent the symbol corresponding to target genes. The more miRNA–target pairs, the larger the miRNA node.

**Table 1 ijms-26-03655-t001:** Summary of small RNA sequencing data generated for 6 samples.

Sample	Raw Reads		3ADT&Length Filter		Junk Reads		Rfam Reads		Valid Reads	
	Total	Uniq	Total	Uniq	Total	Uniq	Total	Uniq	Total	Uniq
P_plus_root1	16,035,908 (100%)	4,620,097 (100%)	3,285,048 (20.49%)	1,748,224 (37.84%)	59,001 (0.37%)	24,745 (0.54%)	1,582,806 (9.87%)	47,073 (1.02%)	11,101,097 (69.23%)	2,799,745 (60.60%)
P_plus_root2	12,653,242 (100%)	4,175,035 (100%)	4,294,605 (33.94%)	2,026,027 (48.53%)	41,521 (0.33%)	18,307 (0.44%)	1,081,337 (8.55%)	35,145 (0.84%)	7,227,792 (57.12%)	2,095,238 (50.18%)
P_plus_root3	17,352,907 (100%)	5,277,691 (100%)	3,184,645 (18.35%)	1,928,169 (36.53%)	80,397 (0.46%)	34,711 (0.66%)	1,379,134 (7.95%)	44,820 (0.85%)	12,703,527 (73.21%)	3,269,733 (61.95%)
P_min_root1	21,267,612 (100%)	5,185,861 (100%)	4,349,634 (20.45%)	2,129,049 (41.05%)	71,407 (0.34%)	26,599 (0.51%)	1,586,045 (7.46%)	31,782 (0.61%)	15,245,191 (71.68%)	2,998,143 (57.81%)
P_min_root2	15,786,575 (100%)	4,599,854 (100%)	4,039,757 (25.59%)	1,967,958 (42.78%)	64,517 (0.41%)	24,490 (0.53%)	1,057,089 (6.7%)	26,580 (0.58%)	10,615,959 (67.25%)	2,580,559 (56.10%)
P_min_root3	19,893,744 (100%)	4,505,317 (100%)	2,746,217 (13.8%)	1,392,144 (30.90%)	83,072 (0.42%)	29,281 (0.65%)	872,518 (4.39%)	21,718 (0.48%)	16,184,397 (81.35%)	3,061,902 (67.96%)

3ATD usually means 3′ Adapter Trimmed Data, which is the data after trimming the 3′ adapter sequence.

**Table 2 ijms-26-03655-t002:** Sequencing results of degradome sequencing.

Sample	DeP_plus (Number)	DeP_plus (Ratio)	DeP_min (Number)	DeP_min (Ratio)	Sum (Number)	Sum (Ratio)
Raw Reads	39,016,365	/	40,529,349	/	79,545,714	/
reads < 15 nt after removing 3 adaptor	213,292	0.55%	234,398	0.58%	447,690	0.56%
Mappable Reads	38,803,073	99.45%	40,294,951	99.42%	79,098,024	99.44%
Unique Raw Reads	9,111,132	/	9,732,496	/	16,259,475	/
Unique reads < 15 nt after removing 3 adaptor	71,443	0.78%	74,632	0.77%	122,517	0.75%
Unique Mappable Reads	9,039,689	99.22%	9,657,864	99.23%	16,136,958	99.25%
Transcript Mapped Reads	29,310,460	75.12%	30,967,313	76.41%	60,277,773	75.78%
Unique Transcript Mapped Reads	5,787,569	63.52%	6,075,096	62.42%	9,648,644	59.34%
Number of input Transcript	638,227	/	638,227	/	638,227	/
Number of Coverd Transcript	163,561	25.63%	167,571	26.26%	187,329	29.35%

**Table 3 ijms-26-03655-t003:** MiRNAs identified in different studies.

This Study	Wan et al., 2012 [22]	Qiu et al., 2015 [23]	Cao et al., 2016 [21]	Deng et al., 2022 [20]	MiRNA Sequence
aly-miR167a-5p	cln-miR167a	-	-	-	UGAAGCUGCCAGCAUGAUCUA
atr-miR164a	cln-miR164a	aly-miR164a	-	-	UGGAGAAGCAGGGCACGUGCA
cln-miR162	cln-miR162d	-	cln-miR162	-	UUGAUAAACCUCUGCAUCCAG
cln-miR164	cln-miR164b	aly-miR164c	-	-	UGGAGAAGCAGGGCACGUGCG
cln-miR6725	cln-miRn1	cln-miR6725	-	-	UGGCAUCUGUCGAGGUCAUCUA
fve-miR397	-	aly-miR397a	-	-	UCAUUGAGUGCAGCGUUGAUG
mdm-miR160a	cln-miR160a	aly-miR160a	-	cln-miR160	UGCCUGGCUCCCUGUAUGCCA
mes-miR167a	cln-miR167f	-	-	-	UGAAGCUGCCAGCAUGAUCUG
mes-miR2111a	cln-miR2111a	-	-	-	UAAUCUGCAUCCUGAGGUUUA
pab-miR156m	cln-miR156i	mtr-miR156g	-	cln-miR156	UUGACAGAAGAUAGAGGGCAC
pab-miR159a_L+1R+1_1ss22CT	-	-	-	cln-miR159c	UUUGGUUUGAAGGGAGCUCUA
pab-miR166f	cln-miR166a	aly-miR166a	-	cln-miR166	UCGGACCAGGCUUCAUUCCCC
pab-miR166f_L+2R-2	cln-miR166l	bdi-miR166f	-	-	UCUCGGACCAGGCUUCAUUCC
pab-miR166f_R-2	cln-miR166c	vvi-miR166a	-	-	UCGGACCAGGCUUCAUUCC
pab-miR390a	cln-miR390a	aly-miR390a	cln-miR390	-	AAGCUCAGGAGGGAUAGCGCC
pab-miR396a-3p	-	-	-	cln-miR396c	CUCAAGAAAGCUGUGGGAAA
pab-miR396b	-	pab-miR396b	cln-miR396a	cln-miR396b	UUCCACGGCUUUCUUGAACUU
pab-miR396g	cln-miR396b	aly-miR396b	-	-	UUCCACAGCUUUCUUGAACUU
pab-miR396g_1ss21TG	cln-miR396a	aau-miR396	-	-	UUCCACAGCUUUCUUGAACUG
pab-miR397a_L+1R-1_1ss15GA	-	-	-	cln-miR23	UUAUUGAGUGCAGCAUUGACG
pab-miR399d	cln-miR399a	aly-miR399b	-	cln-miR399d	UGCCAAAGGAGAGUUGCCCUG
pab-miR482c_1ss11TG	-	pab-miR482c	-	-	TCTTTCCTACGCCTCCCATTCC
pab-miR535a	-	osa-miR535	-	-	UGACAACGAGAGAGAGCACGC
pab-miR536a_1ss20CA	-	-	-	cln-miR536	CCGUGCCAAGCUGCGUGCAAC
ppe-MIR482e-p3_2ss14TC17CT_1	-	-	-	cln-miR48	UCUUGCCUAUUCCCCCUAUGCC
ppt-MIR477d-p5_1ss18TC	-	-	-	cln-miR107	CCUCUCCCUCAAAGGCUCCCA
ppt-miR529e	-	ppt-miR529e	cln-miR529	cln-miR529a	AGAAGAGAGAGAGUACAGCCC
pvu-miR482-3p_L-2R+2_1ss5CG	-	-	-	cln-miR68	UUGCCAAUUCCGCCCAUUCCUA
tcc-miR169m_L-1R+1_1ss2GA	-	mtr-miR169d	-	-	AAGCCAAGGAUGACUUGCCGG
PC-3p-10_276448	-	cln-miR01	-	cln-miR26	UCUUUCCUUUACCACCGAUACC
PC-3p-11729_1020	-	-	-	cln-miR101	AUGUAACAAAGUAAAGCUGCC
PC-3p-1603_5981	-	-	-	cln-miR98	ACGACUGGCAUGUUGAGCACA
PC-3p-1692_5734	-	cln-miR07	-	-	AAUCUAAUGGAAGCCAGUGUU
PC-3p-17_141789	-	-	-	cln-miR92	UUUUCCCUGUACCACCCAUUCC
PC-3p-18_140528	-	cln-miR02	-	-	UACCCAAUGGAUCUUCCCAACU
PC-3p-2_787472	-	-	-	cln-miR08	UUUUCCCUGAACCACCCAUUCC
PC-3p-30807_392	-	-	-	cln-miR72	UCAAUGCUGUACUCAAUAACG
PC-3p-4437_2467	-	-	-	cln-miR75	CCACAUUGAUGAAUUGAUUUC
PC-3p-5_354638	-	-	-	cln-miR50	UAAUGGCUAGUGGUAACUUACC
PC-5p-1647_5857	-	-	-	cln-miR24	UGUUUCUGUUUGUUGACAAUG
PC-5p-17099_712	-	-	-	cln-miR39	AAUCAAUUCAUCAAUGUGGCA
PC-5p-216_29280	-	-	-	cln-miR59	UUUGAGUGAAUCCAGAGUCUCU
PC-5p-25017_487	-	-	-	cln-miR65	UGAAGAGAGAGAGCAUAGCCA
PC-5p-3222_3238	-	-	-	cln-miR46	UCCUCCUAACUGGUGUGAGCUUU
PC-5p-9664_1213	-	-	-	cln-miR51	UUCUUUGCUCUGUUAUGCUCC
PC-5p-970_9048	-	-	-	cln-miR14	UAUAGGGGUAAUGGACAAACU

## Data Availability

Small RNA and degradome data of phosphate deficiency in Chinese fir root have been uploaded to the National Genomics Data Center (Accession number: PRJCA035288).

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
