# Peer review of "Identification of miRNAs and Their Targets in Cunninghamia lanceolata Under Low Phosphorus Stress Based on Small RNA and Degradome Sequencing"

_ijms, 2025, doi:10.3390/ijms26083655_

Round 1
Reviewer 1 Report
Comments and Suggestions for Authors
Overall, this manuscript presents a valuable contribution to the understanding of miRNA-mediated responses to low phosphorus stress in Chinese fir. With more detailed reporting of the methods and improved clarity in the data analysis description, the paper will be significantly strengthened and more accessible to the broader scientific community.
Major Issues:
-
Insufficient Description of Methods:
- The methods section is too terse, making it difficult to evaluate the validity and reproducibility of the results.
- For example, in Figures 6 and 7, a regulatory network is presented, but the manuscript does not adequately describe how this network was inferred. Providing more detail on the computational or statistical approach used for network construction would strengthen the credibility of the analysis.
-
Lack of Detail in Quality Assessment:
- On page 14 (end of Section 5.2), the manuscript states that "the quality of the sequencing data was assessed, and a series of bioinformatics analyses were performed on the raw data" — but there is no further clarification of which specific analyses were performed or how data quality was assessed.
- It would be helpful to include details such as quality thresholds used for trimming, alignment rates, and the handling of low-quality reads.
Minor Issues:
- The manuscript could benefit from more consistent terminology, especially when referring to miRNAs and their target genes.
- A more detailed figure legend for Figures 6 and 7 would improve clarity.
- Some sentences in the discussion are overly long and could be streamlined for better readability.
Recommendations:
- Expand the methods section to provide sufficient detail for reproducibility, particularly regarding network inference and data quality control.
- Clarify the methodology for assessing sequencing quality and bioinformatics analysis.
Reviewer 2 Report
Comments and Suggestions for Authors
In the present study, Li et al. utilized small RNA sequencing of Chinese fir roots to identify micro RNAs that were differentially expressed under low concentrations of phosphorus in the soil. Associated gene regulatory networks that could be activated by phosphorus depletion were also identified.
The work is clearly described and the data are presented in detail. The activation of the above-mentioned gene regulatory networks was only predicted and not experimentally tested. Nevertheless, this work contributes new insights for future experimental studies on the gene expression responses of the Chinese fir to the lack of phosphorus in the soil.
The authors should address the following remarks:
1. In section 2.1, please include a reference for the Solexa technique.
2. Under Table 1, please include a footnote with the full name of 3ADT.
3. In Figures 1, 2, 4, 5, and 7, please label the panels more visibly.
4. In Figures S1, S3, and S4, please label the axes.
5. In Table 3, please identify the references also using their numbers in the reference list.
6. Please re-check the numbering of supplementary tables and figures.
7. In section 5.1, please describe the method in the simple past, for the sake of verb tense consistency with the other methods sections.
8. In the data availability statement, please include the database name.
Author Response
Comments 1: In section 2.1, please include a reference for the Solexa technique.
|
Response 1: Agree.Thank you for pointing this out. We have added a relevant citation for the Solexa technique in Section 2.1 of the Results on page 3, labeled as reference number 24. And the other citation numbers have been adjusted accordingly.
|
Comments 2: Under Table 1, please include a footnote with the full name of 3ADT. |
Response 2: Thank you for pointing this out. We have added a footnote to Table 1: “3ATD usually means 3' Adapter Trimmed Data, which is the data after trimming the 3' adapter sequence.”
Comments 3: In Figures 1, 2, 4, 5, and 7, please label the panels more visibly.
Response 3: We sincerely apologize, due to the small size and compression of the images generated by the software, we have done our best to present them as clearly as possible.
Comments 4: In Figures S1, S3, and S4, please label the axes.
Response 4: Thank you for pointing this out. We have added axis labels to Figures S1, S3, and S4. The revised figures are uploaded as supplementary files.
Comments 5:In Table 3, please identify the references also using their numbers in the reference list.
Response 5: We've added a footnote to Table 3 to clarify the references. Thank you for bringing this to our attention.
Comments 6:Please re-check the numbering of supplementary tables and figures.
Response 6: We apologize for the confusion. Upon checking, we found that Figures S3 and Figure S4 were mislabeled in the manuscript. But we have already revised them in the first paragraph of the Discussion section on page 10. The other supplementary tables and figures are correctly numbered. Thank you for pointing this out.
Comments 7:In section 5.1, please describe the method in the simple past, for the sake of verb tense consistency with the other methods sections.
Response 7: In response to your suggestion, we have changed the tense of Subsection 5.1 in Materials and methods on page 14 to simple past. The revised sentence is: “The "061" clone of Chinese fir seedlings from Fujian Yangkou State-owned Forest Farm (117° 9′ E, 26° 82′ N), with a height of about 25 cm and in good health, were selected. We removed the light substrate and washed the sea sand. And then we divided into normal and low phosphorus groups with 8-10 seedlings each for sand cultivation, and allowed a 7-day recovery period. We prepared a 1/3 strength Hoagland nutrient solution (1 mmol/L KH2PO4 for normal, 0 mmol/L KH2PO4 for low phosphorus with equivalent amounts of KCl supplementing potassium), watered each plant with 100 mL of nutrient solution every 2 days, and treated the two groups for 20 days. Finally, we sampled the roots of Chinese fir seedlings after treatment, with three biological replicates for each treatment, which were immediately frozen in liquid nitrogen and stored at -80℃, and extracted total RNA using the RNeasy® Plant Mini Kit from TIANGEN company. RNA quality and purity were checked using agarose gel electrophoresis and a NanoDrop 2000 spectrophotometer (Thermo, Wilmington, DE) at 260/280 nm (ratio >2.0), and then the samples were sequenced at LC-BIO (Hangzhou, China).”
Comments 8: In the data availability statement, please include the database name.
Response 8: Thank you for pointing this out. We have added the database source in the Data Availability Statement on page 15: Small RNA and degradome data of phosphate deficiency in Chinese fir root have been uploaded to National Genomics Data Center.(Accession number: PRJCA035288). |

Round 2
Reviewer 1 Report
Comments and Suggestions for Authors
The revised version of the manuscript improved the original version in the direction I have suggested.